# OnabotulinumtoxinA Treatment in Chronic Migraine: Investigation of Its Effects on Disability, Headache and Neck Pain Intensity

**DOI:** 10.3390/toxins15010029

**Published:** 2022-12-30

**Authors:** Dilara Onan, Enrico Bentivegna, Paolo Martelletti

**Affiliations:** 1Spine Health Unit, Faculty of Physical Therapy and Rehabilitation, Hacettepe University, Ankara 06230, Türkiye; 2Department of Clinical and Molecular Medicine, Sapienza University, 000189 Rome, Italy; 3Regional Referral Headache Centre, Sant’Andrea Hospital, 000189 Rome, Italy

**Keywords:** OnabotulinumtoxinA, chronic migraine, neck pain, disability, pain intensity, quality of life

## Abstract

Neck disability and pain are frequently encountered problems in patients with chronic migraine (CM). The long-term stimuli of neurons in the trigeminocervical junction may explain this situation. OnabotulinumtoxinA (ONA) treatment is one of the proven treatments for CM; however, there is no study data on the efficacy of ONA treatment on neck disability and pain in CM patients. Therefore, we aimed to investigate the effect of ONA treatment on disability, neck pain and headache intensity in CM patients. One hundred thirty-four patients who met the inclusion criteria were included in the study. ONA treatment was administered at a dose of 195 U to 39 sites in total as per Follow-the-Pain PREEMPT protocol. The disability was evaluated with the Neck Disability Index and the Migraine Disability Assessment; pain intensity was evaluated with the Visual Analogue Scale; the monthly number of headache days were recorded; quality of life was evaluated with the Headache Impact Test. All assessments were recorded at baseline and 3 months after treatment. After the treatment, neck–migraine disabilities decreased from severe to mild for neck and moderate for migraine (*p* < 0.001). Neck pain and headache intensities decreased by almost half (*p* < 0.001). The median number of monthly headache days decreased from 20 days to 6 days (*p* < 0.000). The quality-of-life level decreased significantly from severe to substantial level (*p* < 0.001). According to our results, ONA treatment was effective in reducing neck-related problems in CM patients. Long-term follow-up results may provide researchers with more comprehensive results in terms of the treatment of chronic migraine–neck-related problems.

## 1. Introduction

Migraine is a type of headache that reduces the quality of life and causes disability [1]. Neck pain (NP) is one of the common problems accompanying migraine [2,3,4]. The frequency of migraine attacks is associated with neck disability [5], and according to the World Health Organization, neck pain is among the highest among disability along with migraine headache [6,7].

According to the diagnostic criteria of The International Classification of Headache Disorders-3 (ICHD-3), Episodic Migraine (EM) is defined as having a headache for less than 15 days per month, and Chronic Migraine (CM) is defined as the presence of migraine headaches more than 15 days per month for more than three months, and 8 days per month and more [1]. NP is two times more common in CM compared to EM (OR: 2.04, 95% CI: 1.3–3.3; *p* = 0.008; I^2^ = 10.0%) [5] and CM is associated with neck pain. [5]. In the literature studies, it has been reported that patients experience neck complaints before or during migraine attacks [5,8,9,10,11]. In a meta-analysis published in 2022, the prevalence of NP in migraine patients was reported to be 77%, and NP was found to be twelve times more common in migraine patients than in controls without headache [8].

Neck pain can radiate from the base of the neck to the temporal, frontal and periorbital regions during migraine attacks. Pain may remain permanently in the occipital region, cervical vertebral area and trapezius/deltoid region. Scientific discussion has generally focused on the hypothesis that this complaint arises from the activation of the trigeminocervical system in migraine [12,13]. The nociceptive impulse from the cervical region muscles and dura mater converges on secondary neurons at the trigeminocervical junction. Long-term nociceptive stimuli from the neck region continuously stimulate the nucleus caudal of the trigeminal nerve, causing the trigeminal nerve to become active [10,11,12,13,14,15]. Therefore, it has been stated that cervical nociceptive stimulation may play a role in this activation for migraine headache [13,15]. The cause of neck disability in migraine patients may be due to not using the head and neck region to avoid pain, or it may be due to decreased craniocervical muscle strength-endurance and cervical region mobility as a result of motor cortical problems [16,17,18,19,20]. As a result of these reasons, deterioration in craniocervical posture, such as head forward position, may occur as a compensation [21]. Therefore, neck stabilization cannot be achieved because abnormal loads are placed on the cervical muscles, joints and ligaments [5,22]. This may lead to deficiencies in daily activities that require neck and head stabilization, such as driving, reading and personal care [5,21,22]. Repetitive nociceptive stimulation resulting in disability in daily activities associated with the cervical region may also contribute to the chronicity of pain [23].

ONA is one of the treatments with proven effectiveness in the treatment of CM [24,25,26,27,28]. The European Headache Federation and the Italian chronic migraine group recommended a guide and algorithm for the use of ONA as an effective treatment in CM [27,28]. ONA is administered by injecting a total of 155 U–195 U, 5 U to each region, into between 31 and 39 regions (glabellar, frontal, temporal, occipital, upper cervical and trapezius areas) including the neck and head, every 12 weeks [28,29]. It has been reported that ONA inhibits the calcitonin gene-related peptide, glutamate A and substance P secreted from activated sensory nerves [30,31,32,33,34,35]. It is thought that by inhibiting these inflammatory substances, peripheral sensitization is prevented and central sensitization is also reduced [36].

The disability and pain experienced can affect daily work, family and social life, causing patients to try to find solutions to their problems [29]. The literature studies have shown reductions in migraine-related pain intensity and monthly migraine headache days and improvements in disability and quality of life outcomes with ONA treatment [37,38,39]. However, it is seen that there are no results after ONA treatment related to neck pain and disability in patients with CM and NP. Therefore, in this study, we aimed to investigate how effective ONA treatment is in chronic migraine patients in terms of neck and migraine disability and pain intensity complaints over a 3-month period.

## 2. Results

The mean age of the patients was 53.38 ± 12.36 years. Of the sample, 89.55% were women. The median migraine diagnosis year was 20. The median neck pain duration was 132 months. The demographic and clinic information are given in Table 1.

### 2.1. Primary Outcomes

The Neck Disability Index (NDI) and Migraine Disability Assessment (MIDAS) scores significantly decreased after four weeks of ONA treatment (−16.5 points as median, *p* = 0.000; −28 points as median, *p* = 0.000, respectively). While the patients had severe neck disability before ONA treatment, the level of neck disability was mild after the treatment. The MIDAS level decreased from severe to moderate with ONA treatment (Figure 1 and Table 2).

After treatment, the intensity of neck pain and migraine headaches decreased by almost half (*p* = 0.000; *p* = 0.000, respectively). The median reduction in neck pain intensity was 4 cm, while the median reduction in migraine headache intensity was 5 cm (Figure 1 and Table 2).

The median number of monthly headache days decreased from 20 days to 6 days (*p* = 0.000) (Figure 1 and Table 2).

### 2.2. Secondary Outcome

The HIT-6 score decreased by a median of 10 after treatment (*p* = 0.000). While the quality of life level was severely affected at 68 points before treatment, it decreased to a substantial effect with 58 points after treatment (Figure 1 and Table 2).

## 3. Discussion

This retrospective open-label real-world study investigated the 3-month effects of one session of ONA treatment on neck and headache in patients with CM. Our results showed that in patients diagnosed with CM who experience neck pain and disability, a single session of ONA treatment reduces the disability levels in daily life, neck pain and headache intensity, the number of monthly headache days, and increases their quality of life over 3 months.

The NDI and MIDAS assess disability in daily activities for neck pain and migraine headaches [40,41]. The NDI includes activities such as lifting, reading, sleeping, driving, working, self-care, concentration and recreation [40,41]. The MIDAS evaluates lost time related to work/school, housework, family and social or leisure activities [42,43,44]. Considering the patients’ daily work patterns, housework and leisure activities, it is possible to experience pain-related disability. Effective treatment can reduce the frequency of migraine headaches and reduce the social effects of activities that are disrupted in normal daily life [45]. According to the results of our study, the fact that the patients had a median of 29.5 in the neck disability score and a median of 40 in the headache disability score at the beginning indicated the level of severe disability. This indicated that the patients experienced severe functional limitations in their work life, home, personal and leisure activities due to migraine headache and neck pain. It is likely these restrictions lead patients to seek solutions so that they can continue their daily activities without disability. In our study, in the 3-month results of one session of ONA treatment, the NDI score was reduced ’was reduced to a median of 13, representing mild disability, and the MIDAS score by 12 medians to moderate disability. Therefore, ONA treatment benefits patients by increasing functionality in daily life activities and work activities over a 3-month period. In studies evaluating the effectiveness of ONA treatment at 1, 3, 6 and 9 months, the results are positive for the level of migraine disability [38,46,47,48]. On the other hand, there is no study data evaluating the effectiveness of ONA on neck disability in patients with CM. Mathew et al. reported that the MIDAS score, which was at a severe level with a mean of 34.12 ± 28.93 at the beginning, decreased to a mild level with an average of 10.48 ± 24.09 at the end of the 3rd month with ONA treatment, and patients improved more than 50% (*p* = 0.0541) [47]. Blumenfeld et al., on the other hand, showed an improvement of more than 75% (*p* = 0.0002), with a mean decrease of 21.89 ± 5.78 from the beginning to the 3rd month of ONA treatment [48]. Demiryurek et al. reported that the mean MIDAS score, which was 17.40 at the beginning, decreased significantly to an average of 8.22 at the 3rd month after ONA treatment (*p* < 0.001) [38]. Migraine patients experiencing neck pain are also likely to have reduced cervical spine stability due to the uncomfortable pain and, secondarily, deterioration in muscle tone [5,21,22]. Therefore, it can be predicted that individuals may experience neck-related disability in daily activities [5]. According to our results, the neck-related functionality of individuals with migraine headache who experience loss of productivity and disability [6,49] increased in the 3rd month with ONA treatment.

In our results, we found that with ONA treatment, at the end of 3 months, the median of headache intensity decreased by 50% from 10 to 5 (*p* < 0.001), and the decrease in neck pain intensity was almost 50% (*p* < 0.001). Naderinabi et al. showed that after 3 months of ONA treatment, the intensity of pain decreased significantly from 8.9 cm to 5 cm on average (*p* < 0.001) [50]. Ozon et al. reported that a mean decrease of 2.9 cm was significant in the 3rd month after ONA treatment (*p* < 0.01) [51]. Demiryurek et al. showed that an average decrease of 2.37 cm in pain intensity was significant 3 months after ONA (*p* < 0.001) [38]. Patients with a diagnosis of migraine are more likely to experience NP compared to individuals without migraine [5]. This situation can be explained by the trigeminocervical complex. Migraine patients describe a pain in the frontal part of the head in the ophthalmic cutaneous distribution of the trigeminal nerve. The pain may also radiate to the back and lower part of the head, which is innervated by the occipital nerve, which is a branch of the C2 spinal root, passing through the frontal region. Metabolic activity increases in the trigeminal nucleus caudalis and C1–2 dorsal horn with stimulation of a branch of C2. Prolonged stimuli cause sensitization. These focal neurons are called the trigeminocervical complex. With sensitization involving the head and cervical region, pain, muscle contractions and limitation of movement can be seen. It is stated that the stimulation of the supratentorial dura mater may cause pain by increasing the activity in the trigeminal nerve and upper cervical roots [10,11,12,13,14,15]. Inflammatory substances in activated sensory nerves are inhibited in ONA application and sensitization is reduced [36]. Although there is no study evaluating the effect of ONA treatment on neck pain and disability in patients with chronic migraine who have neck pain, in light of this information, it is likely that the inhibition of sensitization by ONA treatment reduces the intensity of migraine headache as well as the intensity of neck pain.

The decrease of more than 30% in the number of days with headache after the first month with ONA treatment indicates that ONA treatment can be successful [27]. The decrease in monthly headache days in ONA treatment is more than 75% between the beginning and the next application shows that the response to the treatment is excellent [52]. Ornello et al. reported those who responded to a reduction in headache days after ONA application at 3, 6 and 9 months [52]. In the results at 3 months, the rate of excellent responders to ONA treatment was 8.6% (*n* = 248/2879). Since this rate increased in the 6th and 9th months, it was stated that the 3rd-month data were predictive for long-term results [52]. In the results of our study, the median number of days with headache decreased significantly from 20 days to 6 days (*p* < 0.001). In a systematic review of 2022 that synthesized the literature studies, it was emphasized that ONA treatment in migraine headache is an effective option in reducing the number of days with monthly headache, frequency of attacks and the intensity of pain and disability [53]. The review states that ONA treatment is well-tolerated and safe in patients with adult migraine headaches and, therefore, it is successful in reducing headache days [53,54]. Demiryurek et al. reported that the number of days with a headache per month was 18.78 days at the beginning and 12.38 days at the 3rd month after ONA treatment (*p* < 0.001) [38]. Dodick et al. and Silberstein et al. reported that a more than 30% reduction in the number of days with headache in the 3rd month after ONA treatment compared to the baseline was significant (*p* < 0.001) [55,56]. Therefore, we have seen that the results of our study are compatible with the positive and common opinions of the literature on reducing the monthly headache days with ONA treatment.

The intensity of pain and the disability experienced during attacks can cause bed rest and a reduction in work–home–social activities. Therefore, the quality of life of patients with CM is affected [57,58]. In our results, it was significant that the HIT-6 scores decreased from a median of 68 to a median of 58 at the 3rd month after treatment (*p* < 0.001). In our study, we showed similar results with the literature studies, and we found that ONA treatment had positive effects on the quality of life at the 3rd month [59,60]. Results of a population-based study indicated that pain intensity, disability and headache frequency were correlated with quality of life scores (*p* < 0.001) [61]. Beckman et al. reported that decreased headache days after ONA treatment were correlated with increased quality of life in chronic migraine patients [39]. It seems that the decrease in the intensity of headache and neck pain, migraine and neck disability, and the number of days with headaches may increase the quality of life of the patients.

The CM has an economic burden associated with migraine [62]. In a study conducted in a German population with a 2-year follow-up, migraine patients who received ONA treatment every 12 weeks significantly decreased hospitalizations with chronic headache complaints (*p* < 0.02) and visits to health professionals, such as family doctors or migraine professionals (*p* < 0.001) [62]. In real world data, it is stated that ONA treatment is beneficial in the long term by reducing health care-resource usage costs in chronic migraine patients [62]. In another study conducted in Swedish and Norwegian populations, it was stated that ONA treatment increases the quality of life by reducing the monthly headache days in CM patients and reduces the costs related to the disease, and that ONA is a cost-effective treatment method [63]. In a systematic review published in 2022, when the cost-effectiveness ratio of ONA treatment compared to placebo (saline water injection) was examined, ONA was found to be more cost-effective [64].

Some side effects may be reported by patients after ONA administration [65]. Neck pain and neck stiffness are among these side effects. Jackson et al. investigated ONA treatment in adult patients with migraine and tension headache in their meta-analysis study. In the results, they reported neck pain (RR, 4.7; 95% CI, 3.2–6.9) and neck stiffness (RR, 3.2; 95% CI, 1.9–5.6) as adverse effects after ONA administration [66]. Shaterian et al., on the other hand, stated that side effects such as neck pain or stiffness lasted for several days after ONA [53]. Some side effects may be reported by patients after ONA application [65]. In the study of Ahmed et al., neck pain was reported in 2.8% of chronic migraine patients 215 (*n* = 18/633—a mild result of ONA) due to injection in the neck region after ONA treatment [65]. However, since the injection is a stress applied to the tissue, they stated that this may be normal and when the general results are examined, ONA application is safe [65]. In a study, it was reported that the use of a needle as long as 1 inch during ONA application may increase the risk of neck pain side effects [67]. Due to the possible damage to the deeper cervical muscles, the needle length can be considered in the applications in order to avoid side effects by considering the sensitivity of the patients [67]. It is also thought that patients who have had neck pain before may be prone to experience neck pain after ONA application [67]. However, it has been stated that neck pain that occurs with treatment also decreases with repeated ONA treatment [67,68,69]. Therefore, we think that the sensitivity of the patients should be taken into consideration, controlled, and informed in terms of the aggravation of neck pain as a side effect in migraine patients who experience neck pain.

There are several limitations in our study. First of all, the lack of long-term follow-up of the study is one of the limitations. While the literature studies have reported positive results on headache intensity, disability and quality of life for the long-term effect of ONA treatment, such as 6 and 12 months, there are no data on long-term results related to neck problems. Secondly, the psychological status of chronic migraine patients was not evaluated. Since the psychological state can play a role in pain intensity, disability and quality of life, it can affect neck disability and pain intensity. Finally, retrospective studies have some disadvantages. When symptoms related to the disease are very severe, patients may not be able to pay attention at certain points. Due to the imperfect human memory, they may not be able to remember details and may skip some details. Therefore, they may not be able to clearly answer the questions asked. The natural limitations of retrospective studies are that the accuracy of the answers sought during retrospective analysis is uncertain depending on symptom severity and memory, and that different results cannot be analyzed in different time periods because the study is not prospective [70].

## 4. Conclusions

The results of our study showed that headache and disability of patients with chronic migraine can be accompanied by neck disability and neck pain. Since the trigeminocervical junction plays a role in sensitization, it is important to question neck problems in patients with chronic migraine. In addition to the pain and disability in migraine attacks, neck disability and problems related to neck pain may affect the quality of life of individuals and cause deterioration in the daily functional activities of patients. According to the 3-month results of ONA treatment, neck pain, headache and disability experienced by patients with CM decreased and their quality of life increased. However, we think that conducting long-term treatment follow-up studies on neck disability and pain in patients with CM will provide more comprehensive results regarding the management of chronic migraine–neck-related treatment.

## 5. Materials and Methods

### 5.1. Study Design

Ethical approval for this retrospective open-label real-world study was obtained from Sapienza University (52 SA_2020). The study was conducted at Sapienza University Sant’Andrea Hospital Headache Centre with patients diagnosed with chronic migraine by an internist with headache expertise according to the International Classification of Headache Disorders criteria. All patients gave informed consent for the application of ONA treatment. The evaluations of 156 consecutive chronic migraine patients who received one session of ONA treatment in the Headache Clinic between June and October 2022 were analyzed retrospectively with an interview with the patient at the time of the visit for the injection session. Analyzes were performed between September and November 2022. Inclusion criteria were between the ages of 18 and 65 and diagnosed with chronic migraine. Exclusion criteria were diagnosis of any accompanying headache other than chronic migraine, diagnosis of any pathology in the cervical spine, any systemic disease and an acute infection–fracture–inflammatory condition. In total, 134 patients met the inclusion criteria.

### 5.2. Intervention

ONA administration of all patients was performed by an internist with headache expertise. The application included a total of 39 regions as corrugator, procerus, frontalis, temporalis, occipitalis, upper cervical paraspinal muscle group and trapezius. ONA administration was completed by injecting a total of 195 U, 5 Units to each region as per the PREEMPT protocol Follow-the-Pain procedure [27,28,49,71,72].

### 5.3. Outcome Measures

Age, Body Mass Index, migraine-diagnosed years and neck pain duration (months) were recorded for all patients. The primary outcome measures were the Neck Disability Index (NDI), the Migraine Disability Assessment (MIDAS), headache and neck pain intensity and the number of monthly headache days. The secondary outcome measure was the Headache Impact Test (HIT-6). The primary and secondary outcomes were evaluated at baseline and three months after ONA treatment.

### 5.4. The Primary Outcomes

The NDI consists of 10 questions, each scored from 0 to 5, including neck pain, headache, lifting, reading, sleeping, driving, working, self-care, concentration and recreation (0–4 points = no disability, 5–14 points = mild, 14–24 points = moderate, 25–34 points = severe, 35 and above = complete disability) [40,41].

The MIDAS assesses the last 3 months of disability and consists of a 5-item self-administered test, including disability-related in work/school, housework, family and social or leisure activities. The total number of days missed in these activities is the total score and classes the disability as minimal disability (0–5 points), mild disability (6–10 points), moderate disability (11–20 points) or severe disability (≥21 points) [42,43,44]. In addition to the five items mentioned in the test, there are two more items that are not included in the score but provide information to the clinician about the frequency of headaches (MIDAS-A) and pain intensity (MIDAS-B) in the last three months.

The headache and neck pain intensity was assessed with a Visual Analogue Scale (VAS). The patients selected their pain level on a horizontal line between 0 and 10 cm (0 = no pain, 10 = very severe pain) [73].

The number of monthly headache days obtained from the monthly headache diaries of the patients were recorded.

### 5.5. The Secondary Outcome

The HIT-6 is a quality-of-life questionnaire for headaches that assesses vitality, pain, psychological distress, sociability, role and cognitive functioning. Each item is scored on a 5-point Likert scale (6 = never, 8 = rarely, 10 = sometimes, 11 = very often, 13 = always). A score between 36 and 78 points is determined by summing the scores on the six items (≤49 = little/no impact, 50–55 = some effect, 56–59 = substantial effect, and 60–78 = severe effect; a higher score indicates more of a deterioration in quality-of-life) [74,75,76].

### 5.6. Sample Size

The sample size was calculated using G*Power 3 software [77]. The sample size was not calculated at the beginning of the study. Data of 134 patients who came to the clinic for treatment between June and October 2022, according to the inclusion criteria, were collected and post hoc power analysis was performed. The NDI and the MIDAS variables were used in power analysis. For the NDI, the effect size was 0.83, α = 0.05, and the power was obtained as 1.00. For the MIDAS, the effect size was 0.85, α = 0.05, and the power was obtained as 1.00. Since the power calculated for the sample size was sufficient according to the post hoc analysis, the patient recruitment was completed with 134 individuals.

### 5.7. Statistical Analysis

The data analysis was performed using SPSS version 23.0 (IBM Corp, Armonk, NY, USA). The Shapiro–Wilk test of normality analyzed whether the numerical data were normally distributed. The results were given as mean and standard deviation (SD) or median and minimum–maximum for continuous variables; number (*n*) and percentage (%) were presented for categorical variables. The Paired samples *t*-test was used if parametric test assumptions were confirmed, otherwise the Wilcoxon signed-rank test was used to evaluate the difference before and after ONA treatment. Statistical significance was accepted as *p* < 0.05.

## Figures and Tables

**Figure 1 toxins-15-00029-f001:**
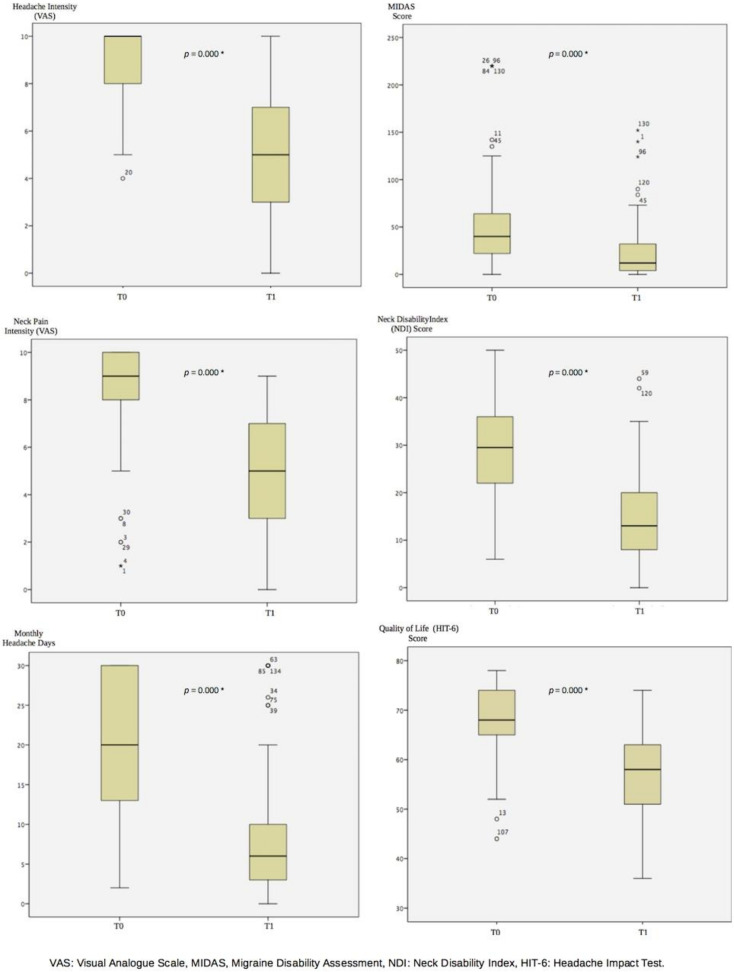
The Results at Baseline and 3 Months After OnabotulinumtoxinA Treatment. T0: Baseline, T1: 3 months After OnabotulinumtoxinA Treatment, *: *p* < 0.001.

**Table 1 toxins-15-00029-t001:** The Demographic and Clinic Information.

Variables	Mean (SD) or Median (Min–Max) or *n* (%)
Age (years)	53.38 ± 12.36
Sex Female Male	120 (89.55%) 14 (10.45%)
BMI (kg/m^2^)	23.9 (14.70–36.60)
Headache Diagnosis (years)	20 (2–48)
Neck Pain Duration (months)	132 (1–552)

BMI: Body Mass Index, SD: Standard Deviation.

**Table 2 toxins-15-00029-t002:** The Results of the Wilcoxon Signed-Rank Test.

Variables	Z	*p*-Value	Effect Size
Migraine Headache Intensity (VAS)	−9.922	0.000 *	0.85
Migraine Disability (MIDAS)	−9.856	0.000 *	0.85
Monthly Headache Days	−9.456	0.000 *	0.81
Neck Pain Intensity (VAS)	−9.351	0.000 *	0.80
Neck Disability (NDI)	−9.665	0.000 *	0.83
Quality of Life (HIT-6)	−9.666	0.000 *	0.85

VAS: Visual Analogue Scale, MIDAS: Migraine Disability Assessment, HIT-6: Headache Impact Test, NDI: Neck Disability Index, ONA: OnabotulinumtoxinA *: *p* < 0.001.

## Data Availability

The data presented in this study are available in thist article.

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
