# Peer review of "OnabotulinumtoxinA Treatment in Chronic Migraine: Investigation of Its Effects on Disability, Headache and Neck Pain Intensity"

_toxins, 2022, doi:10.3390/toxins15010029_

Round 1
Reviewer 1 Report
This is an interesting manuscript but there are several very important issues that the authors need to address before publication can be considered.
The abbreviation used worldwide for many years – and by the company making the product - for onabotulinumtoxinA is ONA. The authors should change the abbreviation they used (OBT-A) to ONA throughout their manuscript. There is no justification for using a new abbreviation.
The text needs editing as there are too many repetitive statements such as “it has been reported….” , ït has been stated….”
The introduction is very mixed. The authors skip between clinical data and toxin science in a very mixed way. I recommend that they revise the Introduction to read much more clearly. As an example, lines 61-66 should be presented much earlier in the Introduction. Other improvements are needed.
The authors should carry out a thorough and more focused literature review for the subject that they address. The statement in lines 164-165 is too direct without this. For example, neck pain in migraine was discussed as long ago as 2012:
Botulinum toxin A for prophylactic treatment of migraine and tension headaches in adults: a meta-analysis
J. L. Jackson, A. Kuriyama and Y. Hayashino
JAMA 2012 Vol. 307 Issue 16 Pages 1736-45
Accession Number: 22535858 DOI: 10.1001/jama.2012.505
The authors have not cited this publication and others similar. Indeed, neck pain is cited as an adverse event when toxin is used for the treatment of migraine, with up to 30% of patients reporting this AE. The authors must discuss how they can study effects of ONA on neck pain when the same condition is an AE. They deal with his AE in line 214 et seq, but only in a cursory way. The authors should pay much more attention to this subject of AE.
Author Response
Dear Reviewer,
Thank you for your comments and contributions.
Response to Reviewer 1 Comments
Point 1: The abbreviation used worldwide for many years – and by the company making the product - for onabotulinumtoxinA is ONA. The authors should change the abbreviation they used (OBT-A) to ONA throughout their manuscript. There is no justification for using a new abbreviation.
Response 1: The OBT-A was edited as ONA in the manuscript.
Point 2: The text needs editing as there are too many repetitive statements such as “it has been reported….” , ït has been stated….”
Response 2: On your suggestion, repetitive statements were edited.
Point 3: The introduction is very mixed. The authors skip between clinical data and toxin science in a very mixed way. I recommend that they revise the Introduction to read much more clearly. As an example, lines 61-66 should be presented much earlier in the Introduction. Other improvements are needed.
Response 3: The Introduction part was rearranged in line with your suggestion. The sentences on line 61-66 were presented before.
Point 4: The authors should carry out a thorough and more focused literature review for the subject that they address. The statement in lines 164-165 is too direct without this. For example, neck pain in migraine was discussed as long ago as 2012:
Botulinum toxin A for prophylactic treatment of migraine and tension headaches in adults: a meta-analysis
- L. Jackson, A. Kuriyama and Y. Hayashino
JAMA 2012 Vol. 307 Issue 16 Pages 1736-45
Accession Number: 22535858 DOI: 10.1001/jama.2012.505
The authors have not cited this publication and others similar. Indeed, neck pain is cited as an adverse event when toxin is used for the treatment of migraine, with up to 30% of patients reporting this AE. The authors must discuss how they can study effects of ONA on neck pain when the same condition is an AE. They deal with his AE in line 214 et seq, but only in a cursory way. The authors should pay much more attention to this subject of AE.
Response 4: "Although there is no study evaluating the effect of ONA OBT-A treatment on neck pain and disability in patients with chronic migraine and neck pain, in the light of this infor-mation we think that the inhibition of sensitization by ONA OBT-A treatment reduces the intensity of migraine headache as well as the intensity of neck pain." In this sentence, we wanted to talk about the effect of ONA treatment in chronic migraine patients with neck pain. Yes, neck pain can be seen as a side effect, but ONA application can also be effective in the treatment of neck pain and disability in chronic migraine patients, and we aimed to investigate this. So we've added a few additions to this sentence to make it clearer.
In line with your suggestion, the article “Botulinum toxin A for prophylactic treatment of migraine and tension headaches in adults: a meta-analysis” was added as a reference.
Paragraph AE was examined in more detail.

Reviewer 2 Report
In this paper, the authors aimed to investigate the effect of OBT-A treatment on disability, neck pain and headache intensity in 134 CM patients and found significant improvements in neck pain and headache intensities, MHD and QOL levels.
The methodology applied is sound and the paper is well written to flow well.
I would only suggest that differences in variables from table 2 can be described in a figure with findings and error bars according to timepoints.
Author Response
Dear Reviewer,
Thank you for your comments and contributions.
Response to Reviewer 2 Comments
Point 1: I would only suggest that differences in variables from table 2 can be described in a figure with findings and error bars according to timepoints.
Response 1: Differences in variables and findings in Table 2 were explained according to time points. However, Table 2 is presented as the Wilcoxon test result.

Round 2
Reviewer 1 Report
The authors have significantly improved this paper in line with my recommendations.